# Virus-like Particle Vaccines: A Prospective Panacea Against an Avian Influenza Panzootic

**DOI:** 10.3390/vaccines8040694

**Published:** 2020-11-19

**Authors:** Nathaniel Nyakaat Ninyio, Kok Lian Ho, Abdul Rahman Omar, Wen Siang Tan, Munir Iqbal, Abdul Razak Mariatulqabtiah

**Affiliations:** 1Department of Microbiology, Faculty of Biotechnology and Biomolecular Sciences, Universiti Putra Malaysia, Serdang 43400, Malaysia; nathanielninyio@kasu.edu.ng (N.N.N.); wstan@upm.edu.my (W.S.T.); 2Department of Microbiology, Faculty of Science, Kaduna State University, Kaduna 800241, Nigeria; 3Department of Pathology, Faculty of Medicine and Health Sciences, Universiti Putra Malaysia, Serdang 43400, Malaysia; klho@upm.edu.my; 4Laboratory of Vaccine and Biomolecules, Institute of Bioscience, Universiti Putra Malaysia, Serdang 43400, Malaysia; aro@upm.edu.my; 5Department of Veterinary Pathology and Microbiology, Faculty of Veterinary Medicine, Universiti Putra Malaysia, Serdang 43400, Malaysia; 6The Pirbright Institute, Woking GU24 0NF, UK; munir.iqbal@pirbright.ac.uk; 7Department of Cell and Molecular Biology, Faculty of Biotechnology and Biomolecular Sciences, Universiti Putra Malaysia, Serdang 43400, Malaysia

**Keywords:** avian influenza, surveillance, avian influenza vaccine, VLP vaccine, M2e, veterinary avian influenza vaccine, universal influenza vaccines

## Abstract

Epizootics of highly pathogenic avian influenza (HPAI) have resulted in the deaths of millions of birds leading to huge financial losses to the poultry industry worldwide. The roles of migratory wild birds in the harbouring, mutation, and transmission of avian influenza viruses (AIVs), and the lack of broad-spectrum prophylactic vaccines present imminent threats of a global panzootic. To prevent this, control measures that include effective AIV surveillance programmes, treatment regimens, and universal vaccines are being developed and analysed for their effectiveness. We reviewed the epidemiology of AIVs with regards to past avian influenza (AI) outbreaks in birds. The AIV surveillance programmes in wild and domestic birds, as well as their roles in AI control were also evaluated. We discussed the limitations of the currently used AI vaccines, which necessitated the development of a universal vaccine. We evaluated the current development of AI vaccines based upon virus-like particles (VLPs), particularly those displaying the matrix-2 ectodomain (M2e) peptide. Finally, we highlighted the prospects of these VLP vaccines as universal vaccines with the potential of preventing an AI panzootic.

## 1. Introduction

The earliest recorded outbreak of highly pathogenic avian influenza (HPAI) in domestic birds was in 1878 in Italy when it was described as ‘fowl plague’ [1]. Since then, several epizootics have occurred leading to the deaths of millions of birds [2]. So far, the emphasis placed on avian influenza (AI) is mostly biased towards the risks that the infection poses to the human population. From the veterinary standpoint, however, AI outbreaks in birds have been severely debilitating to the sustainability of the poultry industry incurring huge financial loses in the poultry industry worldwide [3,4].

AI is caused by various strains of avian influenza viruses (AIVs) that are adapted to avian hosts [5]. Although it is widely considered that the infections with low pathogenic AI (LPAI) viruses (LPAIVs) may only cause mild disease in birds, some LPAIVs have been shown to cause moderate to severe disease. However, these defined disease impacts are largely drawn from experimental infections [6,7]. Highly pathogenic AI (HPAI) viruses (HPAIVs) cause severe and fatal disease with up to 100% mortality in infected poultry [7,8]. Past epizootics such as those that occurred in Southeast Asia [9], China [10], and West Africa [8] had shown that even LPAI viruses such as H9N2 strains caused a mortality rate of up to 60 in affected chicken flocks [11]. The continued prevalence of these viruses in birds is inevitably threatening food security by lowering the production of poultry products for local consumption and export.

Interspecies transmission of AIV remained variable depending on AIV virus strain and the host, birds, and mammalian species. The transmission between domestic and migratory wild birds led to the global spread of H5N8 strains of AIV [12]. The cross-species transmission from birds to mammals such as pigs [13], dogs [14], and humans [15] remained largely in a one-way direction with limited spread from infected mammals to birds [16]. These transmissions did not always cause diseases in wild birds and mammalian hosts but they were important reservoirs that kept AIVs in circulation thereby posing a threat of future outbreaks [17]. Over the past decade, strains of AIV that were first identified in wild birds were reported to cause AI epizootics in certain regions even when contact between the infected domestic avian populations and wild birds could not be established [17,18]. All outbreaks of HPAI in Europe recorded between 1997 and 2005 were caused by strains of AIV that were highly related to strains isolated from wild birds [19]. Interestingly, these viruses were not pathogenic in most of the reservoir hosts, thereby drawing attention to the roles of these wild birds in the evolution, maintenance, and spread of AIVs [20]. The high mutation rates of AIVs and the roles of migratory wild birds in AIV transmissions suggest that the threat of a global panzootic is always imminent. Hence, effective preventive measures such as vaccination and surveillance programmes are essential.

AI surveillance programmes in domestic birds have been widely adopted in countries with histories of AI outbreaks. However, to effectively control future AI outbreaks, active global surveillance of AI in wild birds has recently been prioritised to further understand trends in the mutation and spread of the virus. However, the inconsistency and geographical bias associated with the global surveillance of AI in wild birds have limited the effectiveness of such surveillance programmes [20].

Since the introduction of the H5N2 vaccine in 1992 and the H9N2 vaccine in 1998, widespread administration of AI vaccines against H5N1, H5N2, and H9N2 strains of AIVs to domestic chickens had been employed to control AI outbreaks [5]. These vaccines had been reported to protect poultry birds against the aforementioned viral strains. However, these vaccines are highly strain-/subtype-specific [21]. Due to their narrow spectrum, most of the available AI vaccines are unable to confer protective immunity in immunised animals against other strains of the AIV [22]. Furthermore, concerns have been raised over the inabilities of some strain-specific AI vaccines to completely eradicate enzootic AIVs from areas where these vaccines have been consistently used [23]. Widespread use of the available AI vaccines in domestic birds had also been implicated in the continuous mutation of AIVs and the emergence of vaccine escape mutants [17,24]. These concerns have necessitated the continuous development of AI vaccines. The ideal AI vaccine candidate is a universal vaccine with enhanced immunogenicity, a very broad spectrum, and no mutagenicity [25]. Several studies had shown that AI vaccines based upon virus-like particles (VLPs) displaying the matrix-2 ectodomain (M2e) peptide [26] or the haemagglutinin stalk domain of AIVs [27] had a broader spectrum of immunogenicity than some already commercialised AI vaccines. The M2e sequence is conserved across influenza viruses (IVs), thus it is an important component of universal AI vaccine candidates [28]. In this review, we discuss the current status of AI surveillance and vaccination programmes and highlight the prospects of VLP vaccines as universal AI vaccines.

## 2. Avian Influenza Viruses (AIVs) 

IVs are single-stranded RNA viruses that belong to the family *Orthomyxoviridae* [5]. Of the four types of IVs A, B, C, and D, AIVs belong to type A. Influenza A viruses (IAVs) have birds as their natural reservoirs, hence the terms IAV and AIV are used interchangeably [18,29]. AIVs are capable of infecting mammals and also causing zoonotic infections [2]. The ~13.6 kb negative-sense RNA genome of AIVs comprises eight open reading frames (ORFs), which code for 10 core proteins. These core proteins include three subunits of the IAV polymerase, haemagglutinin (HA), neuraminidase (NA), the M2 ion channel, the matrix protein (M1), and two non-structural proteins (NS1 and NS2) [2]. Two of these proteins, the HA and NA, are employed in the sub-typing of all influenza viruses [30]. Mutations in the hypervariable HA and NA proteins lead to the emergence of newer AIV antigenic variants [31]. Genetic reassortments between different subtypes lead to the emergence of new genotypes and subtypes. These two genetic evolutionary processes lead to an increase in virus fitness to sustain and increase host-range, as well as overcome both innate and acquired immunity [32]. The US Centre for Disease Control and Prevention (CDC) asserts that there are currently 18 and 11 identified HA and NA subtypes, respectively. Except for the H17, H18, N10, and N11 proteins that are specifically found in bat-infecting AIVs, all AIVs containing the other identified combination of HA and NA proteins have been found in avian hosts [33]. 

Predominantly, AIV transmission occurs via direct contact and faecal-oral transmission, but transmission via droplets and aerosols have also been reported [34]. Mutagenic analysis of H5N1 showed that the amino acid asparagine at position 701 of the polymerase basic 2 (PB2) protein was essential for viral transmission [35]. Furthermore, loss of glycosylations at the 158 to 160 amino acid positions of the HA protein of H5N1 is a prerequisite for HA binding to sialylated glycans that facilitates the viral transmission [35]. The NA facilitates the release of AIVs by the cleavage of terminal sialic acids in the virions. Although HA and NA-specific antibodies may interfere with viral pathogenesis, these two proteins are constantly co-evolving to escape the host’s immunity [31]. A correlation between the transmissibility of AIVs and their pathogenicities had been demonstrated [36]. Cui et al. [36] showed a high mortality rate in immunologically naïve SPF chickens that were caged with a chicken infected with the highly pathogenic H5N1. However, no transmission occurred when a chicken infected with the lowly pathogenic H5N1 was caged with immunologically naïve SPF chickens. 

### Pathogenicity of AIVs

AIVs are classified into two pathotypes, LPAIVs and HPAIVs, based on the morbidities and mortalities that result in the infected hosts [37]. The HPAIVs constitute strains of AIVs capable of inducing an “intravenous pathogenicity index” (IVPI) >1.2 or a mortality rate >75% in, at least, eight chickens that are four to eight weeks old within 10 days [38,39]. The most severe epizootics of AI recorded were due to outbreaks of HPAIVs. The most severe outbreak of HPAI recorded was the 2012 outbreak in Mexico, which was caused by the H7N3 subtype resulting in the death of ~4 million poultry birds [38]. In other recorded cases, the H5N1 subtype was the predominant genotype implicated in the HPAI outbreaks in Hong Kong (1997), Bangladesh (2007), Malaysia (2004, 2006, 2007, and 2017), Nigeria (2006 to 2008), and Southeast Asia (2003) [8,9,19,40,41]. The HPAI outbreaks in Mexico (1994), Italy (1997), and USA (2004) were predominantly due to the H5N2 subtype [19]. The H7N3 subtype was implicated in the Pakistan (1994), Australia (1994), Chile (2002), and Canada (2003) outbreaks. The H7N7 genotype was implicated in the 2003 epizootic in the Netherlands [19]. Over the past two decades, the H5N2 and H9N2 subtypes had been implicated in several outbreaks in China [10]. 

The high pathogenicity of AIVs of the H5 and H7 subtypes is believed to be due to the evolution-mediated virulence they had acquired from circulation in poultry [42]. Subtypes of HPAIVs are also phylogenetically classified into two independently evolving lineages, the North American and the Eurasian lineages. The independent evolution of these subtypes is believed to be a result of limited contact between avian species from both continents. Geographical separation of the HPAIV subtypes is further corroborated by phylogenetic analyses of the strains involved in the Hong Kong, China, and Singapore AI outbreaks, which revealed that they belonged to the Eurasian lineage [43,44]. Furthermore, phylogenetic analyses of the nucleoproteins (NP) of H5N1 AIVs isolated from ducks in Siberia revealed that they were homologous to the subtypes isolated from ducks in Hokkaido, Japan. Also, these were phylogenetically homologous to the H5N1 viruses isolated from chickens in Hong Kong and China. All these viral isolates were shown to belong to the Eurasian lineage [45]. AIVs from the North American lineage had been associated with outbreaks of LPAI such as the H7N2 outbreaks in the USA in 1994 and 2002. The H7N7, which also causes LPAI has been shown to acquire enhanced pathogenicity following host adaptation, and AIVs of the H7 subtype have remained in circulation in the USA over the past decade [46]. This may be correlated to the proliferation of HPAIVs of the H7 subtype in the USA. The H10 subtype of the North American lineage was isolated in chickens in Australia in 2010, and also from chickens and ducks there in 2012. Genetic analysis of the H10N7 viruses revealed that they were phylogenetically homologous to isolates that had been in circulation in Australian aquatic birds from 2007 to 2009 [47]. However, wild birds that are susceptible hosts of AIVs are migratory animals. Hence, there is a significant possibility that they can facilitate the spread of AIVs of both lineages to new geographical locations.

As mentioned earlier, various strains of LPAIVs had been shown to acquire enhanced pathogenicity after circulating in the host population over a given period. This acquired host adaptability and enhanced pathogenicity could lead to HPAI in the birds. The evolution of H5 and H7 viruses from LPAIVs to HPAIVs upon circulation in poultry had been linked to the accumulation of basic amino acids in the polybasic motif of the HA cleavage site [32,42]. In a study to ascertain the role of multiple basic amino acids at the polybasic motif of the HA cleavage site as a virulence factor, Suguitan et al. [48] infected BALB/c mice, ferrets, and African green monkeys (AGMs) with two variants of H5N1; the wild-type HPAI A/Vietnam/1203/2004 (H5N1) and a variant that lacked multiple basic amino acids at the polybasic motif of the HA cleavage site. Only the wild type virus elicited lethal viral titres, viral dissemination and elevated titres of proinflammatory cytokines [48]. Other important factors that contribute to the pathogenicity and virulence of AIVs include their ability to adapt to their hosts, their ability to evade the host’s immune response and their ability to infect and adapt to a new host [36]. These factors are mediated by the error-prone nature of the AIV RNA-dependent RNA polymerase (RdRp) and the absence of a proofreading mechanism for error correction in the newly synthesized RNA. This consistently gives rise to viral particles capable of evading the host’s immune responses [36]. Furthermore, when an avian host is co-infected with two or more subtypes of AIV, intracellular re-assortment occurs between these viruses. This is the exchange of entire genomic segments between the co-infecting AIVs within the cells of the infected host [49]. This also plays an important role in the pathogenicity of AIVs by giving rise to viruses with enhanced abilities to evade their hosts’ immunity [30,50].

The ability of AIVs to evolve, acquire enhanced pathogenicity, adapt to their hosts, and evade their hosts’ immune defences and the abilities of their migratory hosts to spread various subtypes of AIVs to new locations present an imminent threat of a global panzootic. Hence, the development and implementation of effective controls and surveillance measures are necessary.

## 3. Surveillance of Avian Influenza Viruses (AIVs)

The measures employed for curbing AI outbreaks and morbidities have played major roles in the journey towards global AI control. Although commendable progress has been made, more effective surveillance programmes, treatment regimens and prophylactic vaccines, which are capable of tackling the rapidly evolving AIV need to be developed. Effective control measures are the best option at preventing an AI panzootic which at present seems inevitable. To achieve this, AIV surveillance serves as an important tool for the understanding of the evolutionary patterns and epidemiology of AIVs.

### 3.1. AIV Surveillance in Wild Birds

AIV surveillance programmes in wild birds are essential because of their role as natural hosts for AIVs. The birds are suitable systems within which AIVs can evolve and circulate [51]. Apart from AIVs of the H17 and H18 subtypes found in bats, surveillance has shown that all other subtypes are perpetuated in wild birds, especially migrating aquatic birds [18,49]. However, the large population sizes and long-distance migrations of wild birds could be limiting factors to effective surveillance. As shown in Figure 1, certain criteria are considered in selecting a target sample population for surveillance, and these include: The population should have a history of AIV infection, the wild bird species should be one in which AIVs are prevalent, and the sample population should be one that is ecologically at risk of AI especially if environmental samples from their habitat are infected with AIVs [52]. 

In March 2013, a novel AIV H7N9 was detected in humans in six provinces and municipalities of China. Subsequently, genomic correlation, coalescent, and phylogenetic analyses were employed to ascertain the origin of the virus and the likely re-assortment events that gave rise to the novel AIV strain. It was established that the *HA* gene of H7N9 AIV originated from duck AIVs, and the *NA* gene was most likely from migratory birds of East Asian origin [53]. Since 2013, there had been five outbreaks of the zoonotic HPAI H7N9 in birds and humans from September 2013 to 2017 although it was asserted that the AIV transmission between species was unidirectional [43]. In March 2019, another outbreak occurred in domestic birds and humans in China, which suggested that the virus had effectively expanded its host range beyond wild birds, and was adapted to both poultry and humans. Genetic analysis of the H7N9 strain implicated revealed significant antigenic drift, which made it necessary to update the available H7N9 vaccine [54]. Interestingly, regular AIV surveillance of poultry-related environments had been embarked upon in China since 2008, whereby 50 to 70 environmental samples were analysed per selected region per year. However, H7N9 was not detected by the regular passive surveillance. The virus was only detected in poultry when the surveillance was reinforced following the detection of H7N9 in a human carrier [54]. While having a regular surveillance programme in place is commendable, it could be improved upon by employing a larger sample size, a more extensive surveillance area, and more frequent sample analyses per year. 

Wild birds are known to maintain most of the various subtypes of AIVs, especially during periods when AI outbreaks in domestic birds are not recorded [25,55,56,57]. Consequently, they have been implicated in the re-emergence of some strains of AIVs after long periods of perceived eradication [20]. A typical case was reported after the 2005 outbreak of HPAI H5N1 in migratory birds in Lake Qinghai, China. In May 2006, another outbreak occurred in the same location, and six AIV strains were isolated from the infected birds and were subjected to whole-genome sequencing. Five out of the six strains were similar to H5N1 A/Cygnus olor/Croatia/1/05 that was implicated in the 2005 H5N1 outbreak in swans in Croatia. The sixth strain was related to the H5N1 A/duck/Novosibirsk/02/05 that was isolated from ducks in Novosibirsk, Russia during the 2005 epizootic [58]. Considering how geographically distant Croatia is from China, it is interesting that five strains of H5N1 genetically similar to a Croatian strain would be found in infected wild birds in China. This indicates how important the role of migratory wild birds is, in AIV transmission. Before 2005, HPAIV surveillance in wild birds was rare. Surveillance studies in wild birds increased rapidly after the outbreak of HPAI H5N1 in several wild bird populations from 2005 to 2006. Most of these more recent studies were targeted at early detections of HPAIVs in wild avian fauna before transmission to domestic birds could occur. Detection of HPAIVs in wild birds should also be taken into consideration in vaccine development and administration. This is because the presence of a HPAIV strain in a given wild bird population is an indication that transmission to domestic birds in that geographical location may occur. Hence, such HPAIV detection should be followed with prophylactic vaccination of the domestic birds that are at risk of being infected.

Until now, the prevalence of HPAIV in wild birds is low, and recorded cases are mostly clustered. This makes it difficult for surveillance programmes targeted at early detection of HPAIVs to effectively achieve their aim of clearly establishing the source of infection during every given HPAIV outbreak [59]. Some surveillance programmes also strive to provide insights on the epidemiology of LPAIVs in wild bird reservoirs. Bergervoet et al. [60] performed a study highlighting the circulation of LPAIVs in wild birds and poultry in the Netherlands from 2006 to 2016. They evaluated strain diversity, geographical distribution, and genetic similarities and differences between wild bird and poultry AIVs. It was established that there was a correlation between host species and the HA and NA subtypes of the infecting AIV. The fact that some LPAIVs detected in wild birds were not found in poultry suggested that the transmission between hosts could be selective, and some viral factors could limit the host range of some LPAIVs. However, the same subtypes of LPAIVs were shown to circulate in poultry, geese, and mallards [55,60]. Interestingly, it was shown that LPAIV infections of ducks were seasonal, and these seasons coincided with the peaks of LPAIV prevalence in wild ducks. It was also highlighted that circulating viruses went undetected for prolonged periods, and reassortment of new LPAIV strains with those in circulation in wild birds occurred frequently [60]. These reassortment events may be responsible for the evolution of virulence from LPAIV to HPAIV. This suggests that surveillance programmes focused on LPAIV are as necessary as their HPAIV-focused counterparts.

Surveillance of AIVs in wild birds is mostly limited to cases where deaths are observed. However, AIVs are mostly non-pathogenic or low pathogenic in wild birds, when compared to poultry birds. Widespread deaths of AIV infected wild birds are relatively uncommon. Prior to HPAI H5N1, the only AI enzootic in wild birds that led to widespread deaths was recorded in South Africa in 1961 following HPAI H5N3 infection of the common terns (*Sterna hirundo*) [51]. AIV surveillance programmes that focus on dead wild birds will not provide sufficient insights into the LPAIV genotypes harboured by wild birds [59]. Consequently, this will limit the preparedness in the event when transmissions to poultry and humans occur. Furthermore, certain instances of AIV surveillance in wild birds appear to be embarked upon during AI outbreaks that spark global concern. Such surveillance programmes are mostly short-lived and provide little insights into the evolutionary trends and transmission cycles of AIVs over time [20,59].

Furthermore, surveillance of AIV in wild birds appears to be more focused on the virus than the hosts. This explains why these avian reservoirs are poorly defined with regards to reservoir species, geographical distribution of these species and the strains of AIV that they predominantly harbour. For most species of wild birds, it is unclear if there is any relationship between viral maintenance and age, sex, migration behaviour, habitat, and feeding behaviour. It has been shown that the avian reservoir species, flock size and flock structure are essential for the maintenance and transmission of AIVs [61]. In-depth surveillance of AIV in wild birds that focuses on the reservoir as much as it does on the AIVs will lead to a clearer understanding of the control strategies to be employed [51]. We believe that the introduction of surveillance programmes that are focused on the reservoir hosts is essential in elucidating the role of these hosts in effective HPAIV transmission.

### 3.2. AIV Surveillance in Domestic Birds

The objectives of AI surveillance in domestic birds vary among programmes. Surveillance programmes in developed economies are mainly focused on how AI affects the trade of poultry products, while those in developing economies are aimed at the effects of AI outbreaks on the development of their poultry industry [62]. Surveillance programmes in domestic birds are also adapted to the areas under study and the available resources. Furthermore, surveillance programmes in poultry have the common limitation of mostly focusing on the AIVs implicated in the most recent epizootics [20]. Consequently, this decreases the preparedness for tackling outbreaks by less recently encountered AIVs. Globally, AI surveillances in domestic birds from 2005 to 2010 were predominantly focused on monitoring the spread of HPAI H5N1 that was reported in over 60 countries in Africa, Asia and Europe [62,63]. HPAI H5N1 in domestic birds remains a global health burden for the severe economic losses incurred by poultry farmers and the threat of zoonotic transmission to humans. Early detection constitutes an important measure in the prevention of devastating AI enzootics [57]. As of 2010, 23 European nations, members of the World Organisation for Animal Health (OIE), had active HPAI and LPAI surveillances in domestic birds. These programmes are ongoing and are either locally funded or funded by the European Union (EU). The administrations of prophylactic vaccines against HPAI during the said period were reported in the Netherlands and Germany while LPAI preventive vaccine administrations were reported in Italy, Portugal, and France [62]. It is likely that less developed economies, especially those without access to EU funding, will be unable to effectively carry out AIV surveillance in domestic birds. This will consequently lower their ability for early detection and control of AI enzootics.

The first outbreak of HPAI H5N1 in Africa was reported in Nigeria in 2006 [64]. Locally performed surveillance studies reported no HPAI H5N1 outbreak from 2007 to mid-2008. However, HPAIV of clade 2.2.1 re-emerged in July 2008 [3]. As reported by the OIE, AI surveillance programmes in poultry in African nations in the 2005 to 2010 period, were performed as four separate projects. The first surveillance programme focused on four nations, the second focused on six nations, the third focused on 15 nations, and the most encompassing surveillance programme was carried out in 47 African nations. The major drawback of these surveillance programmes was that they only lasted from one to three years [62]. More recently, the administration of prophylactic HPAI vaccines to poultry birds in African nations is more widely practised. Despite the administration of prophylactic vaccines, HPAI H5N1 has remained enzootic in poultry birds in Egypt [65]. Since 2011, there has been no report of AI outbreaks involving H5N1 of the subclade 2.2.1.1. However, in 2014, H5N1 A/chicken/Egypt/Fadllah-7/2014 was isolated from symptomatic poultry birds in Egypt, and genetic analysis of the virus revealed ~98% similarity with the H5N1 strain of subclade 2.2.1.1 [66]. More recently, AI surveillance in Egypt from 2016 to 2018 showed that the AIVs of H9N2, H5N8 and H5N1 were still in circulation in domestic chickens and ducks [65]. While this shows the importance of continuous AIV surveillance in poultry, it highlights the need for a universal vaccine as strain-specific AI outbreaks are sporadic, and it is impractical to produce and administer protective vaccines against every AIV strain.

From 2005 to 2009, seven Asian nations reported HPAI outbreaks in poultry. Of these countries, China and Thailand reported having locally funded ongoing surveillance programmes [62]. China, Indonesia, and Vietnam reported having preventive vaccination programmes against HPAI in poultry [62]. In Malaysia, HPAI outbreaks were reported in poultry in 2004, 2006, and 2007. After a decade of no recorded outbreak, a re-emergence was reported in 2017. Inter-district spread of HPAI H5N1 in Malaysia is believed to be due to the smuggling of poultry birds, trading of poultry and poultry products, and infection from wild avian fauna [41]. AI surveillance was carried out in domestic waterfowl from 2007 to 2012 in Bangladesh. It was reported that 4.4% of the tested birds were shown to harbour the AIVs of H1N1, H1N3, H3N2, H3N6, H3N8, H4N1, H4N2, H4N6, H5N1, H5N2, H6N1, H7N9, H9N2, H11N2, H11N3, and H11N6. Interestingly, 99% of the AIV-positive birds were asymptomatic carriers, suggesting that they were vital for maintaining AIVs in circulation, which presented an imminent threat of an outbreak [67]. The effectiveness of bird culling, vaccination, and surveillance as control measures to curb AI transmission in Bangladesh was analysed. The ideal control measure was one that employed all three measures [68]. Furthermore, from the first LPAI H9N2 outbreak in China in 1994 and the first HPAI H5N1 outbreak in 1996, several outbreaks of these AIVs were recorded and they were described as being enzootic in China [69,70]. Also, novel reassortants of AIVs of the H5N6, H7N9, and H10N8 subtypes had been reported to be circulating in poultry birds in China [54,70]. The live animal market practices in China are believed to be the cause of keeping the AIVs in circulation [70]. Hence, consistent surveillance and control programmes are essential as the nation has been described as being geographically conducive for the emergence of novel AIVs [63]. In China, vaccination programmes to curb the spread of AIVs in poultry have been implemented. However, the vaccines have not been reported to completely eradicate the enzootic AIVs [10]. Strict restrictions on the live animal market practices may curtail the AIV circulation in China.

### 3.3. Antiviral Treatment Regimens for Avian Influenza

The first line of control during an AI outbreak is the culling of the infected birds, which is then followed by an antiviral treatment regimen [68]. Current AI treatments include antiviral agents that disrupt viral assembly, attachment, or replication, while other AI regimens act via RNA interference [71,72]. Antiviral agents such as amantadine hydrochloride have been effectively used in HPAIV H5N9-infected turkeys and H5N2-infected chickens. They disrupt viral replication by blocking the AIV M2 ion channel protein [73,74]. Although amantadine is suitable for all bird species, AIVs easily develop resistance to it and the drug residues are found in the meat and eggs of treated birds [72,75]. Some other antiviral agents such as oseltamivir and zanamivir, which inhibit the AIV NA protein, have been effective for AI treatment in chickens [76]. These NA inhibitors are effective against all AIV subtypes [77]. However, a prolonged administration is required for effective treatment of infected birds [72].

Interestingly, the use of herbs in the treatment of AI had also been demonstrated. Essential oils of eucalyptus and peppermint administered to H9N2-infected broilers caused a reduction in microscopic lesions in their trachea, and a significant decrease in mortality [78,79]. Chicken models were also protected after the administration of green tea extracts to birds challenged with H1N1 virus A/NWS/33 [80]. Furthermore, a Chinese herbal medicine, NAS preparation, protected H9N2-infected chickens from exhibiting symptoms. However, the treated chickens were still capable of transmitting the virus to healthy chickens [81]. Some of the studies tested the efficacies of these herbs on AIV-infected cell cultures. Leaf extracts of black currant (*Ribes nigrum folium*) were shown to prevent the entry of H7N7 and H1N1 AIVs into Madin Darby canine kidney (MDCK) cells [82]. Also, *Isatis indigotica* root extracts were shown to prevent viral attachment of H1N1, H3N2, H6N2, and H9N2 AIVs to MDCK cells [83]. 

More recently, avian cytokines had been employed in the treatment of AI. They showed broad-spectrum efficacy against various AIV subtypes and this efficacy was not affected by the antigenic shift of AIVs [72]. Treated birds were also reported to recover rapidly. On the other hand, the cytokines were unstable and as such, were not developed for field applications [77]. Also, the high cost of production was a limiting factor. Short interfering RNAs (siRNA) had been highlighted as a prospective AI treatment [72]. The advantage of siRNAs as a treatment regimen was their specificity to the virus for which they were designed. However, this specificity implied a lack of broad-spectrum for a wide range of AIVs. Also, they were quickly degraded by the host’s immune defence mechanisms [72,77].

## 4. Prevention of a Future Avian Influenza Panzootic

According to the OIE as of August 2018, there are two main AI panzootics; one which started in 2005, peaked in 2006 and progressively resolved in 2012, and the other which has been ongoing since 2013 and peaked in 2015 and 2017 [84]. These events suggested that the present control measures such as AIV surveillance in wild and domestic birds, culling and depopulation of infected birds, and diagnosis and treatment of infected birds may not be sufficient. To control AI outbreaks and potentially prevent a future panzootic, effective vaccines are essential.

### 4.1. AI Vaccines

About 50 years ago, inactivated influenza A vaccines against H5 and H7 AIV subtypes were developed in the USA. These vaccines were first administered to turkeys but they had since been modified and their application had been extended to other bird species and against other AIV subtypes [85]. In the past decade alone, ~108 billion doses of inactivated whole AIV vaccines and ~5 billion doses of live vectored AIV vaccines were administered to poultry, globally [85]. AI vaccines are classified as routine, preventive and emergency vaccines. AI vaccines are considered routine when they are strictly implemented in countries where the AIVs are enzootic. They are preventive when administered in areas where the bird population is at risk of new or re-emerging AI outbreaks. Emergency vaccines are administered in an area with an ongoing AI outbreak [86]. 

Following the H5N2 HPAI epizootic in Mexico that began in 1993, extensive vaccination programmes were implemented in poultry from 1994. This prolonged vaccination led to enhanced herd immunity in poultry in Mexico, and eradication of H5N2 HPAI was achieved as there was no outbreak reported since 1996. However, LPAIVs of H5N2 remained in circulation [87]. In Pakistan, the H7N3 HPAI epizootic of 1995 was also followed by the widespread administration of prophylactic vaccines in poultry. These vaccines were effective in curbing further HPAI H7N3 outbreaks, but AIVs that were genetically similar to the strain involved in the epizootic were isolated in birds up to 2004 [39]. Following the H5N1 HPAI epizootic of 2002 in Hong Kong, poultry birds were vaccinated with a killed H5N2 vaccine. A killed HPAI H5N1 vaccine could not be used because it presented a significant risk to the vaccine production personnel [10,88]. 

In some farms where birds were immunised with the H5N2 killed vaccine, mild outbreaks and a few deaths due to H5N1 were recorded from 9 to 18 days after vaccination. However, no death was recorded after day 18 [88]. This vaccination programme has been extended to other Southeast Asian nations from 2002 until the present [85]. An inactivated whole H5N2; A/turkey/England/N28/1973 vaccine was produced in China and administered for prophylaxis against H5N1 outbreaks [10]. This vaccine was later used in Indonesia, Vietnam, and in African nations such as Egypt [39]. AI vaccine administration has also been approved for zoo birds under strict conditions. This prophylactic measure in zoo birds is essential for their protection and preservation especially with regards to the endangered species. In several member nations of the European Union, prophylactic vaccines against H5N1 HPAIV have been approved for use in free-range poultry and zoo birds [86]. In the Netherlands, a HPAI H7N7 vaccine was administered to zoo birds and the vaccinated birds were shown to develop protective humoral immunity against H7N7 AIV [89]. In order to enhance the immunogenicity of these inactivated vaccines, effective adjuvants had been explored too. Lone et al. [90] demonstrated the correlation between the choice of adjuvants and the abilities of vaccines to provide prophylaxis against HPAIVs in chickens [90]. 

These inactivated vaccines, however, are highly specific and they seldom elicit cross-protective immunity against different strains of AIVs. A mismatch between the available vaccines and the AIV strain in circulation results in high mortality in the unprotected avian population [28]. To surmount this shortcoming, reverse vaccinology [91] and the development of multivalent vaccines were explored [52]. Gomaa et al. [52] developed a trivalent vaccine consisting of inactivated H5N1, H5N8, and H9N2 AIVs. Viral challenge with the H5N1, H5N8, and H9N2 contemporary strains circulating in Egypt confirmed the protective efficacy of the vaccine in chickens [52]. These multivalent vaccines, such as the aforementioned trivalent vaccine developed from AIVs circulating in Egypt, are very effective in conferring prophylaxis against their target strains of AIVs. However, constant updates of such vaccines are often required when the viruses mutate or when there is an outbreak of a different strain or subtype of AIVs. Subsequently, reverse genetics had been employed in the production of AI vaccines harbouring the HA and NA of HPAI H5N1. The prophylactic efficacy of this vaccine was demonstrated in chickens, geese, and ducks [92].

The consistent administration of inactivated AI vaccines has been speculated to spur the emergence of vaccine escape mutants, which may remain enzootic [10]. However, the enzootic H5N1 HPAIV in China, Indonesia, Vietnam, and Egypt was not created by vaccination as the virus was enzootic even before the implementation of the vaccination programme [85]. Vaccine escape AIV mutants may also emerge due to improper or incomplete vaccine administration [85]. Furthermore, it has been reported that previous vaccinations of birds with vaccines against other viruses may affect the efficacy of the AI vaccines administered subsequently [93]. Another important limitation of inactivated whole vaccines is their narrow spectrum. During vaccine production, some AIVs do not grow to the titre required for vaccine production, and the downstream processes required to concentrate the virus are costly. These also limit their suitability for use in killed/inactivated vaccine production [39].

### 4.2. VLP Vaccine Development Technology

Recombinant DNA technology charted a new course in vaccinology when virus-like particles (VLPs) were expressed for use as vaccines. VLPs are self-assembled viral protein complexes whose structures closely mimic those of their parent viruses in conformation and organisation [94]. They are produced by inserting the coding region of the structural protein of a given virus into a plasmid vector followed by transformation or transfection of this recombinant plasmid into the desired expression system [95]. The produced VLPs lack the viral genetic material, hence, are incapable of being infectious and pathogenic [96]. These viral structural proteins utilised in VLP development are usually the capsid or envelop proteins of viruses [94], which give rise to non-enveloped and enveloped VLPs (also known as virosomes), respectively [97]. VLPs of influenza viruses consisting of various combinations of the M1, M2, HA, and NA proteins have been produced and assayed for their immunogenicity [98]. The HA, NA, and M2 ectodomain have been shown to be antigenic and thus, are an important component of AIV VLP vaccines [21,28,98].

When used as vaccines, VLPs are believed to be more immunogenic than subunit vaccines because the latter are prone to incorrect folding, which may diminish their immunogenicity [96]. In comparison to whole inactivated vaccines, VLP vaccines have the advantage of being designed to elicit a more specific immune response based on the antigen used in their production [99]. Advances in genetic engineering have aided the development of VLPs with enhanced immunogenicity. To achieve this, some VLPs are designed to exclude immunosuppressive viral proteins [100] and some are designed as a chimeric fusion of an immunogenic antigen displayed on a nanocarrier [21,28,101]. 

The enhanced immunogenicity of VLP vaccines has been attributed to their ability to interact with key players of the innate immune system, such as dendritic cells (DCs), in a size-dependent manner [102,103] and by VLP interaction with DC membrane receptors [99]. This interaction is initiated by VLP binding with pattern recognition receptors (PRRs) of DCs followed by internalisation, antigen processing and presentation, and induction of adaptive immunity [99]. This interaction of VLPs with PRRs is further enhanced by post-translational modifications (PTM) of the VLPs. Glycosylated VLPs readily interact with glycan recognising receptors of DCs, which also aids VLP uptake [99]. This consequently implies that the choice of an expression system could determine the immunogenicity of the produced VLPs as not all expression systems can perform the desired PTM. Although prokaryotic expression systems are less complicated to use in VLP production, they are incapable of carrying out PTM of the produced VLPs, and these PTM-lacking VLPs have been reported to be less immunogenic than their counterparts expressed in eukaryotic expression systems [101]. Yeast, insect, and mammalian cell lines have been extensively used for the production of AIV VLPs with PTMs [104,105,106] and more recently, AIV VLPs produced in plants have been shown to harbour PTMs, which enhance their immunogenicity [107,108]. 

An important PTM in the immunogenicity of VLPs is glycosylation. Glycan analysis of AIVs revealed several N- and O-glycosylation sites [109], and glycosylation of AIVs has been shown to enhance their virulence [110]. However, for recombinant AIV proteins like the AIV VLPs, this same glycosylation, which enhances the virulence of the wild type AIVs becomes advantageous to enhance the immunogenicity of the AIV VLPs [104]. Recombinant VLPs have been engineered to include inserted glycosylation sites for the production of hyperglycosylated VLPs [109,111]. Although none of the currently approved AI vaccines is produced using the VLP hyperglycosylation technology, the technique bears good prospects for use in the production of highly immunogenic AI vaccines. 

In addition to their ability to perform PTMs, other considerations for selecting a suitable expression system for VLP vaccine production include quality of the produced VLP, yield, ease of purification, cost of production, and vaccine safety (in terms of allergenicity and toxicity). Mammalian expression systems such as HEK239, Vero, CHO-K1, and 239 T cell lines have been shown to produce AIV VLPs with a closer semblance to wild type AIVs in terms of structure and glycosylation pattern [105,112]. Consequently, these AIV VLPs are safe and more immunogenic than their counterparts produced in several other expression systems [113]. However, low VLP yield and high production cost are the major factors mitigating VLP production in mammalian cell lines. 

Ward et al. [108] illustrated the induction of a hypoallergenic immune response when vaccines were administered with N-glycans present on plant-derived AIV VLPs. This suggests that plant-derived AIV VLPs are safe and the PTMs present in the plant VLPs may be safe for use as vaccines. This is in good agreement with the assertion by D’Aoust et al. [114] that plants are an effective expression system for the rapid production of HA-based VLP vaccines, especially in response to a pandemic. Currently, biopharmaceutical firms such as Medicago Inc. (Canada), Caliber Biotherapeutics LLC (USA) and Fraunhofer CMB (USA) have commercially expressed HA-based influenza VLP vaccines in tobacco plant [115]. Furthermore, AIV VLP vaccines expressed in insect cell lines have been shown to be highly immunogenic with a higher VLP yield than mammalian expression systems. However, there are concerns over the allergenicity of insect cell line-produced VLPs, i.e., a baculovirus vector used for VLP production was shown to be contaminated with baculoviruses [105]. This issue can be solved by innovating the approaches for the concentration, capture, purification, and polishing of the specific non-enveloped and enveloped VLPs, to effectively remove the baculovirus residuals [116]. In fact, animal vaccines that were produced using baculovirus expression systems such as Ingelvac^®^ CircoFLEX (Boehringer Ingelheim Vetmedica Inc., Ridgefield, Connecticut, USA) or Porcilis^®^ PCV (Intervet/Schering-Plough, Boxmeer, The Netherlands), have demonstrated their safety and successfully licensed to be used in the Europe and North America [116]. Furthermore, there is a trend towards virus-less gene expression approaches in the field of protein expression in insect cells, in order to tackle the problem of baculovirus contamination of VLPs [117].

#### 4.2.1. AIV Antigens for VLP Vaccine Development

The ideal antigen for use in the development of a highly prophylactic AIV VLP vaccine should be one that meets the criteria of eliciting a robust and broad-spectrum immune response. As earlier stated, antibodies against AIVs target the HA and NA proteins of the viruses [118]. When the HA and NA proteins are used as antigens in VLP vaccine development, the resultant immune response is robust and this humoral immune response particularly targets the globular head of the HA protein. However, the hypervariable nature of the antibody-binding sites on the HA (its globular head) may interfere with the ability of neutralising antibodies to bind to it [119]. Although these HA and NA antigens meet the criterion for inducing a robust antibody production, the antibodies produced do not target a conserved region on the AIVs and hence, are not broadly protective.

On the other hand, the HA stem is a highly conserved antigen, which makes it a good candidate for use in VLP vaccine development against AIVs. However, the immunodominance of the globular head has been shown to mask the immunogenicity of the stem protein [120]. To overcome this challenge, headless HA stems have been used in VLP vaccine development. Although the globular head of the HA was initially thought to be essential for HA binding to DC PRRs and cellular uptake, more recent evidence showed that the HA stem alone is capable of such an interaction [121]. Furthermore, the HA stem was shown to be weakly immunogenic, however the humoral immune response elicited was broadly neutralising against different strains of AIVs as compared to antibodies induced by the whole HA antigen [100,119].

The M2e antigen is another conserved domain of AIVs, which is a suitable candidate for use in the development of a broadly protective AIV VLP vaccines. Analyses of the immunogenicity of the M2e peptide show that it induces a protective humoral immune response against AIV that is not based on antibody-mediated neutralization, but rather acts through antibody dependent cytotoxicity (ADCC), as well as other arms of the immune response [122]. However, when compared to killed/inactivated AIV vaccines, the M2e peptide induces a less robust immune response. Its weak immunogenicity, however, remains a major challenge for its usage as a highly prophylactic vaccine [123]. 

To enhance the immunogenicity of these conserved AIV peptides that show good prospects for use as a universal AI vaccine, structurally optimised adjuvants [124], and the epitope display technique of VLP production are widely used [21,125]. Here, the genes that code for peptide epitopes such as the HA stem or the M2e are fused to the genes of other viral structural proteins and inserted in a suitable plasmid vector. The encoded chimeric protein is then expressed and typically, these expressed proteins assemble into chimeric VLPs where the structural protein component acts as a nano-carrier displaying the antigenic AIV epitope [119,125]. The displayed antigens in these chimeric VLPs have been shown to induce the production of significant antibody titres that are capable to protect the immunized animals [28]. Interestingly, an AIV vaccine that combined M2e and HA VLPs was highly immunogenic [126]. This suggests that a combination of the antigens may meet the essential criteria of robust and broad-spectrum immune response induction. 

#### 4.2.2. AI VLP Vaccines

VLPs are empty and non-pathogenic viral capsid proteins. Although they do not contain the genetic materials of the parent virus, they are structurally similar. In some cases, VLPs have been engineered to display foreign antigenic epitopes and peptides [101]. VLP-based vaccines have already been licensed for human and veterinary use, and many other vaccine candidates are undergoing different stages of evaluation [127]. They have been shown to be more immunogenic than other vaccine candidates due to their efficient uptake by DCs and macrophages [99,128], their self-adjuvanting property [129], and their ability to elicit cellular immunity by inducing size-dependent uptake by DCs [86,102,103,130]. In vivo analysis of universal VLP vaccine candidates against AI had shown good prospects. Lee et al. [118] developed a VLP vaccine expressing the HA and M1 of H9N2. Administration of this vaccine to chickens elicited a robust and protective immune response even after viral challenge with wild type H9N2 [118]. 

Various VLPs have been used as carrier proteins for the insertion of antigenic AIV epitopes and these chimeric VLPs are highly immunogenic and cross-protective [21]. Commonly used AIV epitopes in vaccine development are the highly conserved M2e peptide [21] and/or the conserved headless stalk of the HA peptide [100]. The M2e is a 23 amino acid-long type III transmembrane tetrameric protein, which acts as a pH-regulated proton channel. M2e is a highly conserved component of the AIV M2 protein. The M2 protein is essential for AIV replication and infection. Hence, it exists in all AIVs and is also conserved across all AIV subtypes [131,132]. Since its discovery in 1981 by Lamb et al. [133], attempts have been made to develop an M2e-based universal influenza vaccine. Also, M2e-based vaccines had been shown to decrease AIV titre [134], restrict viral growth, and decrease morbidity in AIV-challenged animal models [108,109]. However, unlike the inactivated whole AI vaccines, the M2e is weakly antigenic, and thus the M2e-based vaccines do not lead to viral clearance in vaccinated subjects [107,110]. 

Conversely, chimeric VLPs consisting of a carrier protein displaying the M2e peptide have been shown to enhance its immunogenicity thereby resulting in the induction of protective antibody titres and cellular immunity [135]. Epitope display enhances the presentation of the antigenic epitope to the immune system thereby generating a more robust immune response [50,111]. Furthermore, it has been shown that there is a correlation between the size of VLPs and their immunogenicity. DCs show a size-dependent preference during VLP uptake [102,103,130]. To develop more immunogenic AI, the M2e peptide of IAVs had been displayed on VLPs of hepatitis B virus core antigen (HBcAg) [136], the capsid protein of *Macrobrachium rosenbergii* nodavirus (MrNV) [21,28], the P particle of norovirus [22,137], human papillomavirus L1 protein [138], the coat proteins of papaya mosaic virus [139], tobacco mosaic virus [140], AP205 RNA bacteriophage coat protein [141], infectious bursal disease virus (IBDV) capsid protein [142], and rabbit hemorrhagic disease virus (RHDV) capsid protein [143]. As illustrated in Figure 2, these chimeric VLPs showed good prospects for use as universal AI vaccines.

On the other hand, immune responses elicited by the AIV’s HA are directed towards the hypervariable globular head, which in turn, masks the immunogenicity of the conserved HA stalk [100]. In the first immunological analysis of the HA stalk, it was shown that they elicited a cross protective immune response against AIVs [144]. This makes the HA stalk a suitable epitope for use in the development of chimeric VLP vaccines against AI. 

### 4.3. VLP Vaccines as Universal Avian Influenza Vaccines

According to the US National Institute of Allergy and Infectious Diseases (NIAID), the definition of a truly universal influenza vaccine is that it has at least 75% efficacy against symptomatic infection by group A (influenza A) and group B (influenza B) viruses [25]. Also, the prophylactic efficacy of a universal influenza vaccine must last for at least a year in the vaccinated subjects [145]. More emphasis is placed on the development of universal influenza A vaccines than other groups of influenza viruses since they cause more virulent zoonotic infection [25].

In an early effort towards creating an epitope-based universal AI vaccine, Graves et al. [144] demonstrated that the stalk component of the HA elicited a cross protective humoral immune response against AIVs. The HA stalk, however, was shown to be poorly immunogenic [146]. To surmount this problem, Steel et al. [27] constructed chimeric VLPs consisting of headless HA and linker proteins. This vaccine was shown to elicit the production of neutralising antibodies in mice. Furthermore, the vaccine was shown to protect mice that were challenged with the PR8 strain of AIV. The protective efficacy of headless HA-based VLP vaccines in chickens is yet to be extensively studied.

On the other hand, in the earliest development of M2e-based AI vaccines, Neirynck et al. [147] fused the *M2e* gene to the *HBcAg* gene and they expressed the chimeric vaccine candidate in *E. coli*. The vaccine candidate conferred a 90–100% protection on mice challenged with the AIV, H3N2 [147]. The same vaccine candidate was subsequently shown to completely protect mice against the AIVs A/Victoria/3/75 (H3N2) and A/Puerto Rico/8/34 (H1N1) [136]. The M2e peptide had also been displayed on VLPs of MrNV capsid protein [21] where the chimeric VLPs were shown to completely protect mice challenged with H1N1 and H3N2 IAVs [28]. 

Elaish et al. [137] developed a universal AI vaccine for chickens by displaying the M2e epitope on the P particle of norovirus. The vaccinated chickens were challenged with H5N2, H6N2, and H7N2. The vaccine induced a robust IgY immune response although it was administered without adjuvants. This further demonstrated the self-adjuvancy of VLP vaccines and the good prospects of M2e-based VLPs as universal AI vaccines [137]. Furthermore, M2e displayed on the tobacco mosaic virus coat protein was expressed in tobacco plants. The VLPs were also shown to protect mice challenged with H1N1 [140]. Chimeric VLPs of M2e displayed on the coat protein of papaya mosaic virus were also shown to protect mice against lethal doses of H1N1 [139]. Similarly, the L1 protein of human papilloma was shown to be a suitable carrier to display IAV M2e [138]. In addition, IBDV VP2 VLPs were shown to be able to display different forms of HA, as well as M2e, inducing efficient protection against AIV and offering the possibility of generating bivalent vaccines against two important avian viral diseases: IBD and AIV [142]; while the study reporting chimeric RHDV-VLPs displaying M2e epitope evaluated the influence of foreign-epitope insertion site within the VLP structure over the immune response and protection elicited, which was shown to be higher when the epitope was displayed within a highly exposed loop at the surface of the VLPs [143].

The mechanism of M2e-based VLP vaccines conferred cross-protection in vaccinated animals and humans had been described. Although the M2e-specific antibodies were shown to be incapable of direct neutralisation of AIVs, they induced antibody and cell-mediated immune mechanisms such as antibody-mediated cytotoxicity, antibody-mediated cellular phagocytosis, and complement-mediated cytotoxicity [25]. Interactions between the Fc receptors of DCs, macrophages, neutrophils, the Fc region of the M2e antibodies, and virus-infected cells resulted in the recognition, killing, or phagocytosis of AIV infected cells [148,149,150,151]. Also, these M2e-VLPs triggered the killing of AIV infected cells by triggering the complement cascade, which was achieved by complement binding to the Fc region of the M2e antibodies. This then led to the formation of a membrane attack complex and then lysis of the infected cells [152]. Furthermore, M2e vaccines triggered T-cell responses, and these responses were believed to be crucial for the cross-protection induced by these vaccines. However, the mechanism of T-cell induction by M2e vaccines is still unclear [25]. 

The major concern surrounding the use of M2e as universal influenza vaccines is their low immunogenicity. However, this concern is alleviated by the role of the carrier VLPs in enhancing the immunogenicity of M2e epitopes [103,130]. Also, to enhance the efficacy of these universal AI vaccine candidates, they are either displayed multiple copies of the M2e epitope, administered with adjuvants or co-administered with other influenza vaccines [153]. The M2e based VLP vaccines also meet the criterion of safety due to their non-replicative nature [145]. Thus, they represent a safe and broad-spectrum class of vaccines with good prospects for preventing an AI panzootic.

### 4.4. Limitations and Challenges Associated with VLP Vaccines

As vaccine candidates, VLPs possess various biophysical properties that are advantageous with regards to their ability to induce a prophylactic immune response in vaccinees. However, certain factors relating to the VLP composition and the production process, their safety as vaccines and their stability may be disadvantageous. 

An important drawback of VLP vaccines is the ability of certain components of the VLP to suppress the immunogenic antigen through a process known as carrier induced epitopic suppression (CIES) [154]. This specifically occurs with chimeric VLPs. When the vaccinated host produces antibodies to the nanocarriers used to display the antigenic epitope then such a response could suppress the immune response directed at the displayed epitope [154]. This challenge may be surmounted by using carrier proteins of viruses incapable of infecting the vaccinated subjects [101]. Even then, if this virus has an immunogenic region that is homologous to that found in a virus to which the host is susceptible, antibody production may still be induced [155] thereby resulting in CIES.

In addition, factors relating to the expression system used in VLP production may pose a challenge to the intended use of the VLPs as vaccines. Bacterial expression of VLPs may create the risk of endotoxin contamination of the produced VLPs, which will require elaborate purification to make the VLP vaccines safe for use [156]. For VLPs expressed in insect cells, there are concerns that the baculoviral contaminants could suppress the immunogenicity of the antigenic epitope of the vaccines [157]. Overall, purification of VLPs could involve several steps of density gradient ultracentrifugation and chromatography. Therefore, large-scale production of VLP vaccines could be capital intensive, which will consequently give rise to expensive vaccines [158].

Since the VLP structure is essential to its immunogenicity, problems such as incorrect protein folding and assembly may jeopardise the VLP’s prophylactic efficacy [157]. It has also been reported that fusion of an antigenic epitope to a carrier protein may destabilise the VLP structure by interfering with protein folding [99]. Finally, because of their morphological similarity to viruses, VLPs administered as vaccines may be sequestered in cellular endosomes after uptake [159]. This will interfere with the desired trafficking of the VLPs within immune cells and as a consequence, the immunogenicity of the VLP vaccine may be suppressed.

### 4.5. Current Trends in AIV VLP Vaccine Development

The current state of VLP vaccine development features efforts directed at designing VLPs with enhanced abilities to interact with PRRs on DCs, and VLPs designed to induce more robust and specific immune responses. With the increasing understanding of the interaction between DC PRRs and VLPs, VLP uptake can be enhanced by engineering them to contain peptide sequences than foster enhanced interaction with DC PRRs [99]. The effective antigen processing and presentation of DCs in comparison to other players of the innate immune system makes them a good target to focus on with regards to VLP design. In addition, to enhance the immunogenicity of VLP vaccines, newer design methods that include proteins with adjuvanting properties are also explored [125]. In an earlier reported study, Salmonella flagellin protein was incorporated into VLPs harbouring AIV HA and M1 proteins. These VLPs were reported to induce more robust antibody and cytokine production than their counterpart, which did not contain the flagellin [160]. Also, certain carrier proteins have been identified to exhibit adjuvanting properties [95]. Therefore, their utilisation for displaying AIV antigenic epitopes gives rise to vaccines that do not require adjuvants to induce a protective immune response. One of such carrier proteins is the capsid protein of MrNv. Ong et al. [28] showed that this capsid protein displaying three copies of the IAV M2e antigen protected mice challenged with H1N1 and H3N2 IAVs even in the absence of adjuvants.

More recently, bioinformatic design algorithms for *de novo* proteins are being developed for the creation of functional antigenic epitopes. These epitopes are designed to contain complex structural motifs that facilitate the induction of neutralising antibodies with enhanced antigen-binding specificity [161,162]. VLP vaccines produced using this novel technique induced a neutralising immune response in mice and non-human primates against the respiratory syncytial virus [162]. This technique shows good prospects for use in VLP vaccine development against other viruses such as AIVs.

## 5. Conclusions

HPAIV outbreaks have led to severe loses to the poultry industry worldwide. The available treatment regimens, surveillance programmes, and inactivated or killed vaccines have been shown to be effective in some cases. However, AIVs rapidly mutate and the available prophylactic vaccines, which are subtype-specific, require frequent updates to be effective against new strains of AIVs. The shortcomings of the currently available prophylactic vaccines, surveillance programmes, and treatment regimens suggest that a panzootic is imminent unless more effective control measures are developed and employed. The most pertinent control measure required is a universal vaccine capable of protecting birds against all AIVs. Chimeric VLP vaccines consisting of M2e and/or headless HA have shown the most prospect in this regard due to their broad spectrum and cross protective immunogenicity.

## Figures and Tables

**Figure 1 vaccines-08-00694-f001:**
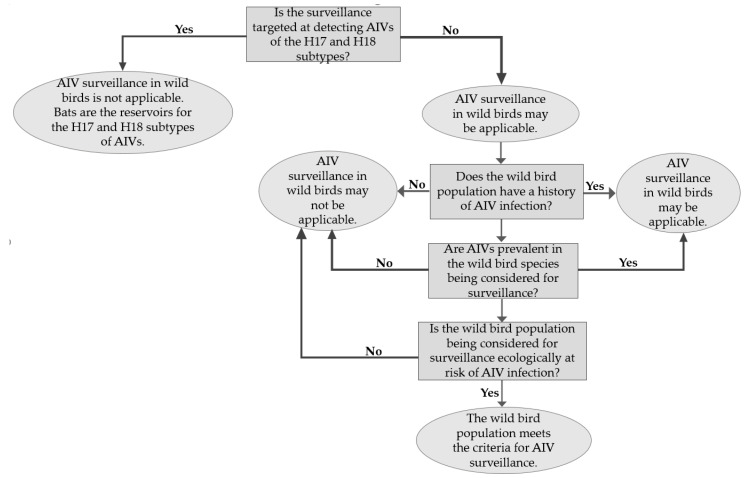
A flowchart representation of the criteria for selecting a wild bird population for avian influenza virus (AIV) surveillance.

**Figure 2 vaccines-08-00694-f002:**
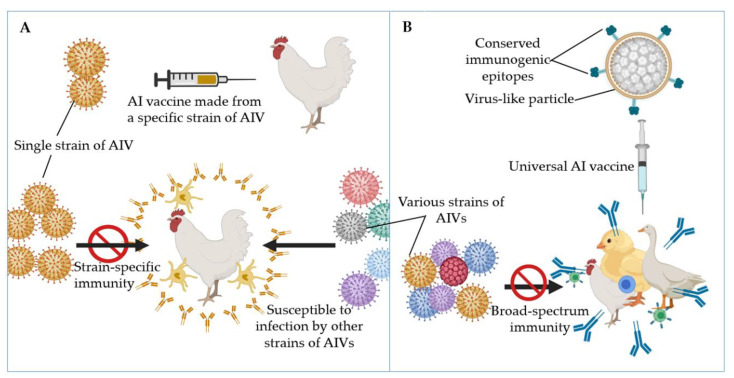
Immunogenicity of strain-specific avian influenza (AI) vaccines versus universal AI vaccines. (**A**) Vaccines derived from specific strain(s) of avian influenza viruses (AIVs) confer strain-specific immunity in vaccinated animals. (**B**) Universal AI vaccines harbouring M2e and/or headless haemagglutinin (HA) stalk confer broad-spectrum immunity in vaccinated animals. This is because the M2e and HA stalk components of these vaccines are conserved across all strains of AIVs.

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
