# Peer review of "Virus-like Particle Vaccines: A Prospective Panacea Against an Avian Influenza Panzootic"

_vaccines, 2020, doi:10.3390/vaccines8040694_

Round 1
Reviewer 1 Report
Highly pathogenic avian influenza (HPAI) has serious impacts on poultry industries world wide and studies of preventing measures including surveillances and vaccine developments are important. Virus-like particle (VLP) vaccines is one of the key technologies regarding vaccine development in medicine and veterinary medicine. However, despite the current title, the authors' focus in this paper is rather comprehensive explanation about avian influenza viruses (AIVs) and surveillance efforts against it, and relatively little space is spent for description of AIV vaccines and VLP. In fact, the authors only spent less than three pages out of total of twelve pages in the text for description of the VLP vaccine, despite it is the major topic in the title. Also, regarding the VLP vaccine, its technological details were not described at all.
In conclusion, although this review paper includes some useful knowledge, overall it does not have consistent focus throughout the paper, which reduces its significance in the field. My suggestion would be to focus on the VLP technology and choices of AIV antigens, including more description about the technology and its advances, if the authors would keep their current title. Or, I would suggest to change the title, and make more general review paper about description of prevention measurements against AIV. In any case, I do not recommend to publish this paper on the journal with the current form.
Author Response
Many thanks for the comment. We have edited the manuscript to include a new section (4.2) entitled “VLP Vaccine Development Technology”. In this section, we highlighted the techniques used in the development of VLP vaccines (lines 444 to 462). We also described the molecular attributes of VLP vaccines that account for their enhanced immunogenicity (lines 463 to 485). Furthermore, we discussed the factors that should be considered in the selection of a suitable expression system for VLP vaccines (lines 486 to 505).
In subsection 4.2.1, we discussed the various AIV antigens utilised in the development of VLP vaccines against AIVs. We also compared the ability of the various antigens to elicit a robust and broad spectrum immune response (lines 507 to 530). In lines 531 to 541, we highlighted the various techniques used in enhancing the immunogenicity of some of these AIV antigens used in VLP vaccine development.
Finally, we included a new section (section 4.5) where we discussed the current advances in AIV VLP vaccine development (lines 676 to 702). We highlighted current trends in designing VLP vaccines to enhance their interaction with dendritic cells. We also discussed new techniques in designing VLPs with incorporated proteins that have adjuvanting properties. In addition, we highlighted current bioinformatics techniques employed for VLP vaccine design that allows for more specific immune response induction.
Therefore, we request to keep the current title of the paper.
Reviewer 2 Report
This manuscript by Ninyio et al. is a comprehensive review of AIV surveillance, treatment regimens, and universal vaccines. It is well written and organized review. A minor concern is that the review emphasized the limits of conventional vaccine; however, the cons of VLPs is not elborated. The disadvantages of VLPs would be better to be included in the revision.
Author Response
Many thanks for the comment. We included a section (4.4) where we highlighted the limitations of VLPs as vaccine candidates (lines 650 to 678). In this section, we highlighted limitations regarding the safety of certain VLPs as immunogens. We also discussed the impact that incorrect VLP assembly may have on their immunogenicity and their potential sequestering by cellular endosomes after uptake.